# Structural basis for catalysis of human choline/ethanolamine phosphotransferase 1

Zhenhua Wang[1,2], Meng Yang[1,2], Yufan Yang[1,2], Yonglin He[1] & Hongwu Qian [1] ✉

Phosphatidylcholine (PC) and phosphatidylethanolamine (PE) are two primary components of the eukaryotic membrane and play essential roles in the maintenance of membrane integrity, lipid droplet biogenesis, autophagosome formation, and lipoprotein formation and secretion. Choline/ethanolamine phosphotransferase 1 (CEPT1) catalyzes the last step of the biosynthesis of PC and PE in the Kennedy pathway by transferring the substituted phosphate group from CDP-choline/ethanolamine to diacylglycerol. Here, we present the cryo-EM structures of human CEPT1 and its complex with CDP-choline at resolutions of 3.7 Å and 3.8 Å, respectively. CEPT1 is a dimer with 10 transmembrane segments (TMs) in each protomer. TMs 1-6 constitute a conserved catalytic domain with an interior hydrophobic chamber accommodating a PC-like density. Structural observations and biochemical characterizations suggest that the hydrophobic chamber coordinates the acyl tails during the catalytic process. The PC-like density disappears in the structure of the complex with CDP-choline, suggesting a potential substrate-triggered product release mechanism.

Phosphatidylcholine (PC) and phosphatidylethanolamine (PE) are the most abundant glycerophospholipids in eukaryotic cells, comprising approximately 50% and 20% of the total cellular phospholipid mass, respectively[1]. PC and PE are primary components of the eukaryotic membrane and play essential roles in maintaining membrane integrity[2], membrane fusion[3], lipid droplet biogenesis[4], autophagosome formation[5], cytokinesis[6], lipoprotein formation and secretion[7,8], and regulating the function of membrane proteins. Metabolic disorders of PC and PE are linked to lipodystrophy[9], fatty liver disease[9], insulin resistance[10], lung surfactant deficiency[11], and some cancers[12,13]. In most mammalian cells, PC and PE are synthesized via the Kennedy pathway[14], the final steps of which are catalyzed by three integrated membrane proteins, choline phosphotransferase 1 (CPT1), choline/ethanolamine phosphotransferase 1 (CEPT1) and ethanolamine phosphotransferase 1 (EPT1). CPT1 catalyzes the displacement of CMP from CDP-choline by diacylglycerol (DAG) to produce PC[15], while EPT1 catalyzes an analogous reaction to produce PE using CDP-ethanolamine and DAG[16]. CEPT1

has dual specificity to use both CDP-choline and CDP-ethanolamine to produce PC and PE, respectively[17].

CPT1, CEPT1 and EPT1 belong to the CDP-alcohol phosphotransferase (CDP-AP) family, whose members share a common property of catalyzing the transfer of a substituted phosphate group from a CDP-linked donor to an acceptor alcohol. CDP-APs possess a conserved signature motif $D_1xxD_2GxxAR...GxxxD_3xxxD_4$ to coordinate the CDP moiety[18], designated the CDP-binding motif. The four conserved aspartates constitute a di-metal binding site to coordinate two catalytically important divalent metal ions[19]. Apart from PC and PE, CDP-APs also catalyze the final biosynthetic steps of other phospholipids in both eukaryotes and prokaryotes, including phosphatidylinositol (PI), cardiolipin (CL), and phosphatidylglycerophosphate (PGP). According to the hydrophilicity of the substrates, CDP-APs are classified into three subgroups, I, II, and III. Most prokaryotic phospholipid synthases belong to class I, which process a lipophilic CDP-linked donor and a polar acceptor, while eukaryotic choline and ethanolamine phosphotransferases (CPT1, CEPT1 and EPT1) are classified as class II CDP-APs

[1]The First Affiliated Hospital of USTC, MOE Key Laboratory for Membraneless Organelles and Cellular Dynamics, Hefei National Research Center for Interdisciplinary Sciences at the Microscale, Division of Life Sciences and Medicine, University of Science and Technology of China, Hefei 230001, China. [2]These authors contributed equally: Zhenhua Wang, Meng Yang, Yufan Yang. ✉e-mail: hongwuqian@ustc.edu.cn

because they utilize a polar CDP-linked donor (CDP-choline/ethanolamine) and a lipophilic acceptor (DAG) as their substrates. Cardiolipin synthases represent the third class, whose CDP-linked donor and acceptor are both lipophilic[18].

Structural information on the CDP-AP family is limited to class-I homologs from bacteria and archaea[18–23], impeding the elucidation of the catalytic mechanism for these eukaryotic CDP-AP members. Here, we report the structures of full-length (FL) human CEPT1 (hCEPT1), a eukaryotic class-II CDP-AP, and its complex with CDP-choline at overall resolutions of 3.7 Å and 3.8 Å, respectively. An in vitro assay was also established to measure the activity of CEPT1 variants. Structure-guided biochemical characterizations indicate the presence of a hydrophobic chamber to accommodate the acyl tails of lipids during the catalytic process.

## Results

### Structural determination of CEPT1
Details for the recombinant expression and purification of hCEPT1 are described in the methods (Fig. 1a). The enzymatic activity of the purified protein was confirmed by a high-performance liquid chromatography (HPLC)-based assay. In detail, the purified protein was mixed with diacylglycerol (DAG)-containing 1-palmitoyl-2-oleoyl phosphatidylcholine (POPC) micelles following a reported protocol[24], and the reaction was initiated by the addition of CDP-choline or CDP-ethanolamine. The release of CMP was quantified using HPLC and validated by mass spectrometry (MS) in tandem with HPLC (Supplementary Fig. 1a). CMP was released under the reported conditions[24] in the presence of wild-type enzymes, the substrate DAG and the cofactor $Mg^{2+}$ (Supplementary Fig. 1b–d). CMP release was approximately doubled when CDP-choline was used as the substrate compared with

CDP-ethanolamine, consistent with previous reports[17] (Supplementary Fig. 1e). Therefore, CDP-choline was added as the substrate in all assays described hereafter. Substitution of one conserved aspartate in the CDP-binding motif with alanine (D154A) completely abolished the enzymatic activity. No CMP release was detected in the absence of DAG, confirming that CMP was not produced by the hydrolysis of CDP-choline (Fig. 1b). Consistent with previous reports, the enzymatic activity was nearly abolished when other divalent cations were added instead of $Mg^{2+}$, except $Mn^{2+}$; with the addition of $Mn^{2+}$, the enzyme maintained approximately 50% of the activity[17] (Fig. 1c). The binding affinity of the enzyme with CDP-choline was 46 μM as measured by isothermal titration calorimetry (ITC) (Supplementary Fig. 1f); this result was comparable with the reported $K_m$ value[24].

To elucidate the catalytic basis of CEPT1, we determined the structure of hCEPT1. Enzymes purified in 0.005% GDN yielded homogenous particles suitable for cryo-EM analysis (Supplementary Fig. 2a). A representative 2D average indicates that hCEPT1 may be purified as a dimer (Supplementary Fig. 2a); therefore, C2 symmetry was attempted during data processing. Eventually, the structure of hCEPT1 was determined to be 3.7 Å resolution using 459,021 particles (Fig. 1d, Supplementary Fig. 2, 3a, Supplementary Table 1). These particles also resulted in a similar 3D map at a slightly poor resolution when applying C1 symmetry, supporting the application of C2 symmetry (Supplementary Fig. 3a).

A dimeric model was built based on the cryo-EM density, of which 380 side chains (28-407) were assigned for each protomer (Fig. 1e, Supplementary Fig. 2d, 3b). Ten transmembrane helices (TM1-10) were observed in each protomer, among which TM5 and TM9 were shorter. Preceding the first TM1 are two N-terminal helices (NH1 and NH2) and one transverse helix (TH) that lies along the cytosolic boundary of the

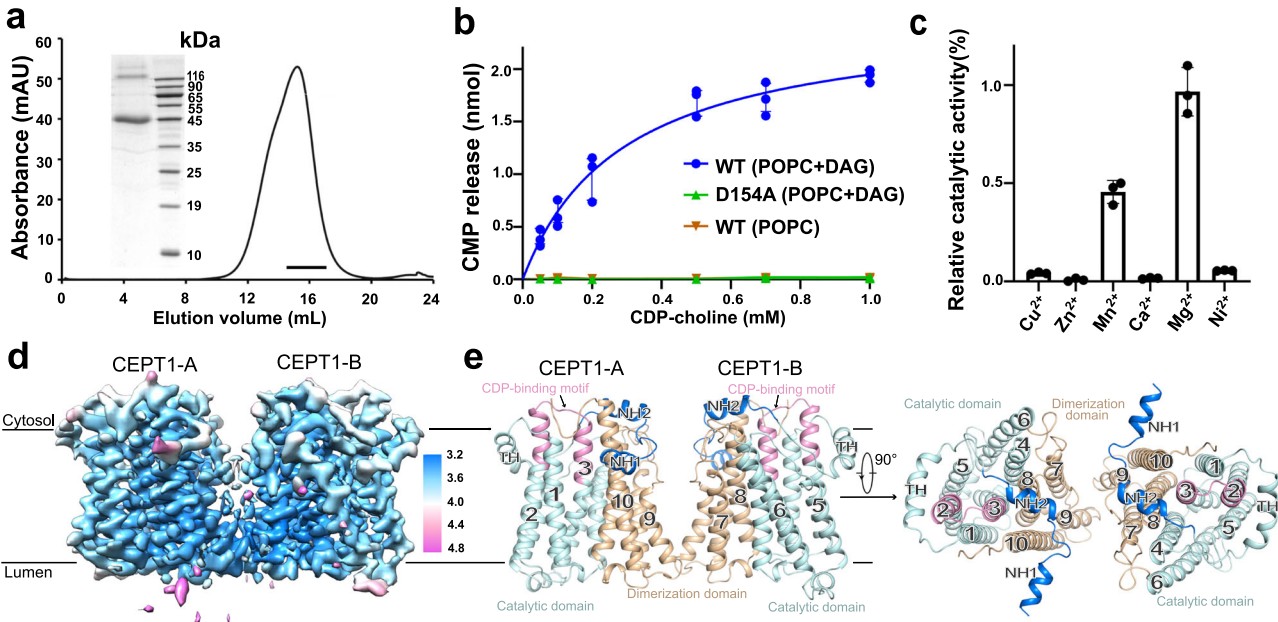

**Fig. 1 | Cryo-EM structure of human CEPT1. a** Representative size exclusion chromatography profile of CEPT1 solubilized with 0.005% GDN. The peak fractions (15–17 ml) were concentrated for cryo-sample preparation. Inset: SDS–PAGE analysis of the concentrated protein used for cryo-EM sample preparation. The experiments were independently repeated more than three times with similar results. **b** CMP release in 10 min catalyzed by wild-type (WT) enzyme in the presence and absence of diacylglycerol (DAG), and inactive variant (D154A). CMP was separated with high-performance liquid chromatography (HPLC) and quantified with peak area. See Methods and Supplementary Figure 1 for details of the assay. **c** hCEPT1 uses $Mg^{2+}$ as the cofactor to produce PC. $Mg^{2+}$ activated the production of PC by hCEPT1, and $Mn^{2+}$ showed approximately 50% efficiency compared with $Mg^{2+}$. CMP release was

nearly undetectable when using $Cu^{2+}$, $Zn^{2+}$, $Ca^{2+}$, and $Ni^{2+}$ as cofactors. Data in **b** and **c** are the mean ± s.d. of three independent experiments. **d** Cryo-EM structure of hCEPT1 was determined as a dimer at an overall resolution of 3.7 Å. A local resolution map was calculated using cryoSPARC[36] and generated with Chimera[40]. The resolution bar on the right is labeled in Å. The two protomers in the dimeric structure are labeled CEPT1-A and CEPT1-B. **e** The dimeric structure of hCEPT1 is exhibited in two perpendicular views. Protomers of the dimer are domain-colored. The catalytic domain and dimerization domain are colored pale cyan and wheat, respectively. N-terminal helices (NH) are colored blue, and CDP-binding motifs are colored pink. The same color code is applied to all figures unless otherwise indicated. TH: transverse helix. All structural figures were prepared with PyMol[44].

membrane (Fig. 1e, Supplementary Fig. 4a). Structural comparisons with bacterial/archaeal homologs[20,21,23] suggest a conserved catalytic domain consisting of TM1-6 (Fig. 1e, Supplementary Fig. 4b–d). Distinct from the bacterial/archaeal homologs dimerized by direct interactions between TMs 3/4/6 of two catalytic domains[20,21,23], the dimerization of CEPT1 is mediated by a transmembrane domain consisting of TMs 7-10, designated the dimerization domain (Fig. 1e). The dimerization domain is superimposed with TMs 3-6 of the catalytic domain with r.m.s.d. of 5.5 Å over 107 Cα atoms, resulting in a C2 pseudosymmetry axis at their interface (Supplementary Fig. 4e, f). These results indicate that the CEPT1 protomer may be derived from a dimeric ancestor, in which one catalytic unit lost its enzymatic activity and became the dimerization domain during evolution.

Dimerization of CEPT1 is mainly mediated by van der Waals interactions of TMs 7/9 at the luminal side of the dimerization domains. In detail, Met302 in TM7 and Phe358/Leu359/Tyr362/Phe363 in TM9 primarily constitute the hydrophobic interface between the two dimerization domains, among which Phe358 may form π-π interactions with Tyr362 in another protomer. Distinct from the tight interaction at the luminal leaflet, the cytosolic leaflet is separated with no direct interaction between the protomers. However, some extra densities were sandwiched by hydrophobic residues and a cluster of four histidine residues from two protomers at the cytosolic leaflet (Supplementary Fig. 5a). This observation indicates that some amphipathic molecules (endogenous lipids or exogenous detergents) may be involved in dimer formation, and their nonpolar tails and polar heads may be stabilized by hydrophobic residues and histidine residues, respectively. The enzymatic activities were not affected when introducing mutations to the dimeric interface (Supplementary Fig. 5b), indicating that the two protomers may catalyze the reaction independently.

## Coordination of a PC molecule in the hydrophobic chamber

In each CEPT1 protomer, the CDP-binding motif is located at the cytosolic side of TMs 2 and 3, and four conserved aspartates point their carboxyl groups to a polar cavity surrounded by TMs 2/3/5 and TH, suggesting catalysis within the cavity (Supplementary Fig. 6a, b). CDP-choline/ethanolamine can enter the catalytic cavity liberally from the cytosol. This observation is consistent with previous studies on bacterial/archaeal homologs[18–23]. However, the bottom of the catalytic cavity is open to a deep hydrophobic chamber, which is distinct from the bacterial/archaeal homologs (Supplementary Fig. 6b). A lipid-like density was unambiguously observed in the structure of CEPT1, of which acyl tails reside in the hydrophobic chamber and the polar head is placed in the catalytic cavity (Fig. 2a). Because no lipid was added during protein purification or sample preparation, the density may indicate an endogenous lipid. Initially, we attempted to model a DAG molecule, the substrate of the enzyme, into the density. However, a portion connecting the 3-OH group remains void, which is enough to accommodate a phosphocholine or phosphoethanolamine moiety. This observation suggests that the density belongs to a PC or PE molecule, the products of CEPT1. Both PC and PE match well with the density, and we modeled a PC molecule into the density to facilitate the description. After fitting a PC molecule, the phosphate group is close to the carboxyl groups of four conserved aspartates on the CDP-binding motif, which coordinate two metal ions to stabilize the pyrophosphate of the CDP moiety in the bacterial homologs[19,20]. CEPT1 prefers $Mg^{2+}$ for PC production[17]; therefore, we modeled two $Mg^{2+}$ ions at similar positions to stabilize the phosphate group of PC in the structure (Fig. 2b, Supplementary Fig. 3b, and Supplementary Table 1).

The product PC is surrounded by TMs 2-6. Two tails are coordinated in the hydrophobic chamber with hydrophobic residues, including Thr162/Val165/Val166/Thr169 of TM3, Cys185/Met192/Phe193 of TM4, Val213/Val216/Phe219/Ile220 of TM5, and Cys253/Thr254 of TM6 (Fig. 2c). Among them, V216A was reported to reduce the catalytic activities and alter the profile of DAG utilization[25]. The

activity reduction was confirmed with our functional assay, which showed a 40% loss of activity for the variant V216L (Fig. 2d). We also mutated several other small residues to bulky residues to limit the space for accommodation of the tails, including T162F, T169N, and A196L, which led to a 50% ~ 70% loss of activity compared with the wild-type enzyme (Fig. 2d). In addition, Gln217 seems to form a hydrogen bond with the carbonyl oxygen of the sn−1 acyl group (Fig. 2c). The activity was reduced by 60% after the hydrogen bond was disrupted by the substitution of Gln217 with leucine (Fig. 2d).

The head group of PC is coordinated by TMs 2/3/5 in the catalytic cavity. The negative charges of the phosphate group are stabilized by two $Mg^{2+}$ ions coordinated by four carboxyl groups from conserved aspartate residues of the CDP-binding motif. Mutational analysis revealed that the activities of the variants D136A and D154A were nearly abolished, and the activity of the CEPT1 (D158A) mutant was approximately 15% that of the wild type (Fig. 2c, e). The trimethylammonium group may interact with the carboxyl group of Glu65 electrostatically. Supporting this observation, the CEPT1 (E65A) mutant showed an approximately 30% loss of activity (Fig. 2c, e).

Apart from connecting with the catalytic cavity, the hydrophobic chamber opens to the center of the membrane through a cleft between TM5 and TM6, where one tail of PC is exposed (Supplementary Fig. 6c). This observation suggests that the PC product may leave the chamber when the cleft is enlarged. Consistent with this hypothesis, several polar residues ride at the top of the chamber, including Asp212/Ser261/Tyr265 (Supplementary Fig. 6c), which may coordinate the polar head of PC when it moves in the chamber. To support this hypothesis, the enzymatic activities were decreased when Asp212, Ser261 and Tyr265 were replaced by asparagine, alanine and phenylalanine, respectively, to reduce or remove the polarity of these residues (Fig. 2f).

## Coordination of CDP-choline

To elucidate the catalytic mechanism, coordination information between enzymes and substrates is needed. We, therefore, incubated purified enzyme with 10 mM CDP-choline and then prepared cryo-samples. Eventually, we obtained a 3D map at 3.8 Å resolution (Supplementary Fig. 7, 8, and Supplementary Table 1). The overall structure is almost identical to our first structure with r.m.s.d. of 0.4 Å over 368 Cα atoms (Fig. 3a, Supplementary Fig. 9a). However, the lipid-shaped density disappeared in the hydrophobic chamber, and a new density arose in the catalytic cavity (Fig. 3a). The CDP-choline molecule matches well with the new density (Fig. 3b, Supplementary Fig. 8b). This observation suggests that CDP-choline binding may trigger the exit of PC from the hydrophobic chamber and leave space for another substrate, DAG.

CDP-choline is primarily coordinated by the CDP-binding motif. The cytidine group is sandwiched between TM2 and TM3 and potentially forms hydrogen bonds with Asn86 and Glu151 (Fig. 3c). Substitution of Asn86 with alanine led to an approximately 80% reduction in activity, and approximately 10% activity was reserved when Glu151 was mutated to alanine (Fig. 3d). Minor shifts of TMs 2/3 were observed to leave efficient space to accommodate the cytidine moiety (Supplementary Fig. 9b). The pyrophosphate moiety is stabilized by the two metal ions that are coordinated by four conserved aspartate residues from the CDP-binding motif. The trimethylammonium group of choline may also be stabilized by Glu65, similar to what we observed in the PC coordination (Fig. 3c).

## Discussion

CDP-APs, which catalyze the transfer of a substituted phosphate group from a CDP-linked donor to an acceptor alcohol, are critical players in phospholipid biosynthesis. Here, we present cryo-EM structures of hCEPT1, a eukaryotic CDP-AP, and its complex with CDP-choline. The CEPT1 structure is a dimer, which is distinct from the monomeric model predicted by AlphaFold2[26]. Moreover, our structures exhibit the

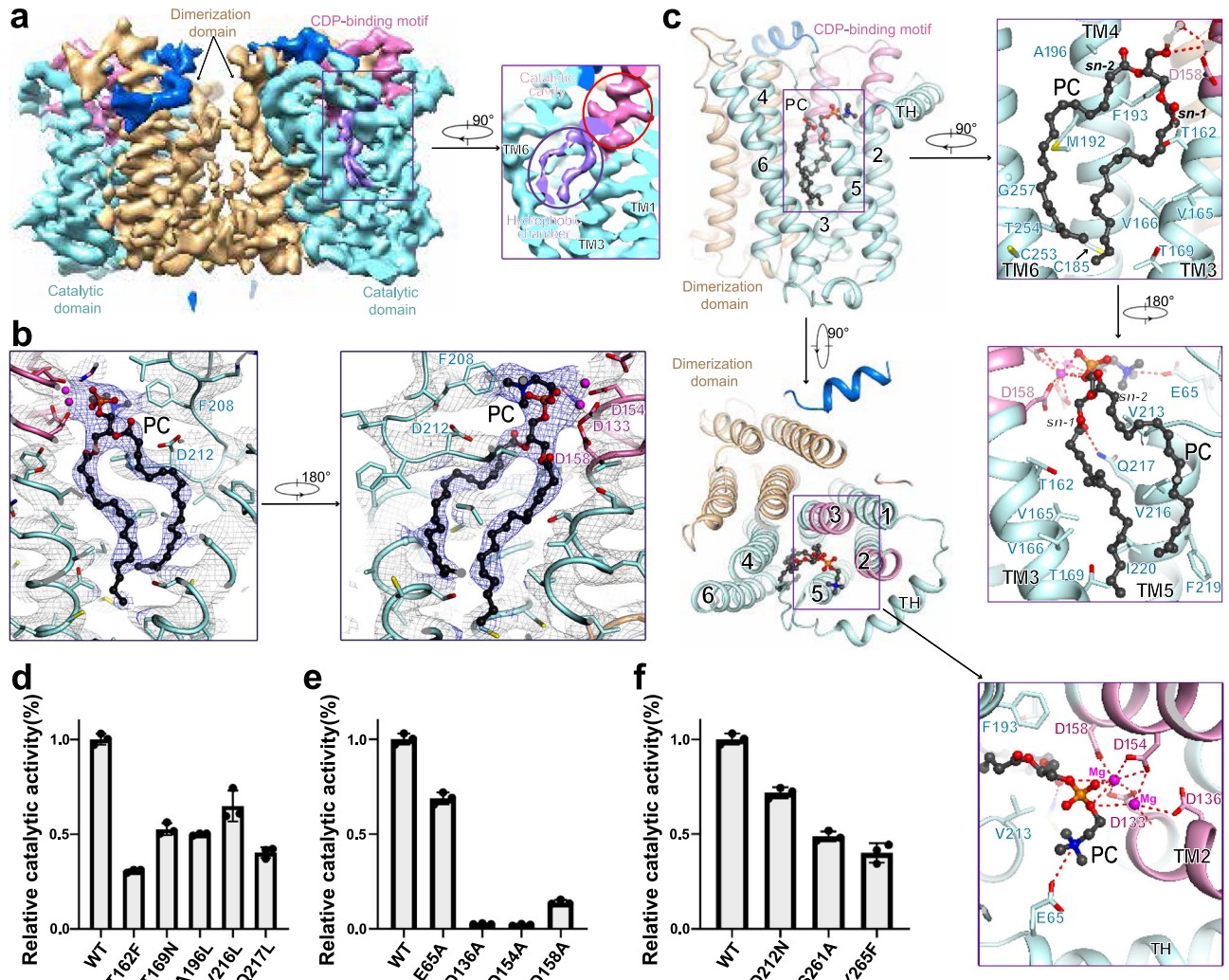

**Fig. 2 | A hydrophobic chamber accommodates an endogenous phosphatidylcholine in CEPT1. a** A lipid-like density is observed in the hydrophobic chamber (purple circle) and catalytic cavity (red circle) in each protomer. The density, shown as purple mesh, is contoured at 5.5 σ. **b** The product PC fits well with the lipid-like density. The lipid densities are contoured at 5.5 σ and presented in two opposite views. **c** The coordination of the endogenous PC. Left: two perpendicular views are presented to show the relative position of PC in the catalytic domain. Right: coordination of two acyl tails of PC in the hydrophobic chamber in two opposite views (upper and middle) and that of the hydrophilic head of PC in the catalytic

cavity in the cytosolic view (bottom). The PC molecule, colored black, is shown as balls and sticks. Two Mg²⁺ ions are shown as magenta balls, and the potential electrostatic interactions are indicated by dotted red lines. **d, e** Functional validation of the importance of the residues that engage in the coordination of PC. **f** Functional verification of the importance of the polar residues on the top of the hydrophobic chamber. The activities of the mutants in **d, e,** and **f** were normalized relative to that of the wild type. Data in **d, e,** and **f** are the mean±s.d. of three independent experiments.

coordination of product PC and substrate CDP-choline in hCEPT1. Combined with biochemical characterizations, these structural observations promote the understanding of the catalytic process of CEPT1.

In the structure of CEPT1, the catalytic cavity opens to the intramembrane via a deep hydrophobic chamber (Supplementary Fig. 6c). Two tails of an endogenous PC molecule were accommodated in the hydrophobic chamber and its polar head was placed in the catalytic cavity (Fig. 2a), suggesting that the hydrophobic chamber may coordinate the acyl tails of lipids during the catalytic cycle. The PC-like density disappeared in the complex structure of CEPT1 and CDP-choline (Fig. 3a), which may suggest that substrate binding is required for the release of the product. Because the complex structure of CEPT1 and CDP-choline was solved under conditions with a vast excess of substrate (10 mM CDP-choline vs. 0.3 mM enzymes), this speculation requires biochemical validation under physiological conditions in the future.

We propose a catalytic model for CEPT1. CDP-choline can enter the catalytic site from the cytosol freely. CDP-choline binding triggers

the release of PC molecules produced during the latest catalytic cycle, emptying the hydrophobic chamber to accommodate another substrate, DAG (state A). Then, DAG enters the hydrophobic chamber from the cleft between TMs 5 and 6 and places its 3-OH group close to the catalytic site (state B). After the reaction, the product CMP leaves from the cytosolic side of the catalytic cavity directly. The product PC may stay in the catalytic chamber (state C) until a new CDP-choline molecule enters the catalytic side to start the next catalytic cycle (Fig. 4). Our structures of CEPT1 and its complex with CDP-choline may represent state C and state A, respectively. The existence of state B requires more structural and biochemical evidences, especially the complex structure between CEPT1 and DAG.

It was suggested that the β-phosphorus atom of the CDP-linked donor could be nucleophilically attacked by the hydroxyl group of the acceptor alcohol, which may be deprotonated by a nearby basic residue[20]. Two basic residues, His155 and His197, are close to the glycerol backbone of the endogenous PC molecule in our structure (Supplementary Fig. 9c). Histidine can function as a general base to

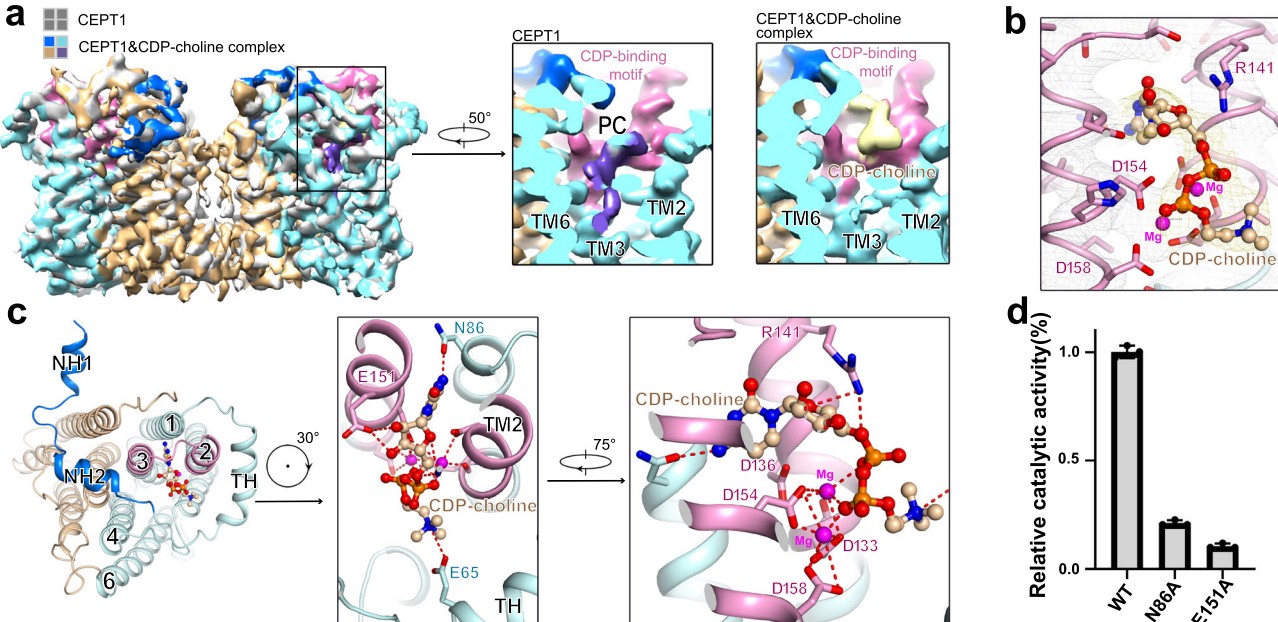

**Fig. 3 | Coordination of CDP-choline in the catalytic cavity. a** Comparison of the EM maps of CEPT1 (silver) and its complex with CDP-choline (CEPT1&CDP-choline complex, domain-colored). Insets: the extra densities in the hydrophobic chamber and catalytic cavity of EM maps of CEPT1 (left) and its CDP-choline complexes (right). The density corresponding to the PC molecule largely disappeared in the complex map of CEPT1 and CDP-choline. The CDP-choline density is colored yellow. **b** The CDP-choline density is contoured at 5 σ. **c** The coordination of CDP-

choline in the catalytic cavity. The CDP-choline is mainly surrounded by TMs 1-3 (left), and the details of coordination are shown in two views (middle, right). CDP-choline is shown as balls and sticks, and the potential electrostatic interactions are indicated by dotted red lines. **d** Functional verification of the importance of the residues in coordination with CDP-choline. The activities of the mutants were normalized to that of the wild type. Data are the mean±s.d. of three independent experiments.

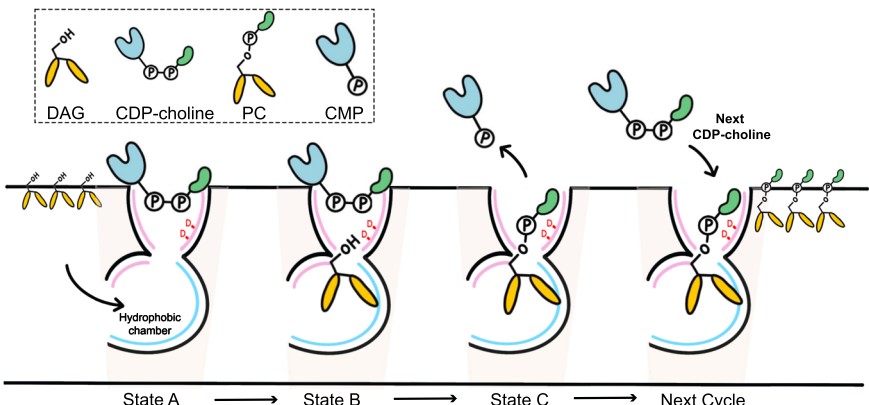

**Fig. 4 | Working model of CEPT1.** The catalytic cycle is proposed to be divided into three states, A, B and C, representing CDP-choline bound, CDP-choline and DAG bound states before reaction, and PC bound state after reaction, respectively. At

the end of one catalytic cycle, a new CDP-choline may be required to trigger the release of product PC, and then the next cycle starts.

deprotonate and activate nucleophilic substrates in several enzymes[27–29]. Therefore, we propose that the 3-OH group is deprotonated by His155 or His197 when DAG enters the catalytic site. Then, deprotonated DAG attacks the β-phosphorus atom of CDP-choline, and the product PC is produced (Supplementary Fig. 9c). The complex structure of CEPT1 with DAG is required to distinguish which histidine acts as the primary base to activate the 3-OH group.

A hydrophobic groove is observed between the catalytic domain and dimerization domain to accommodate some discontinuous densities, and a phospholipid is coordinated at the corresponding position of a bacterial homolog[23] (Supplementary Fig. 9d, e). This indicates that a phospholipid may bind to the hydrophobic groove to stabilize the structure of CEPT1. Consistent with this, PC and PE were reported to be activators of CEPT1[24].

Although more structural and biochemical investigations are needed to answer the remaining questions, our research on CEPT1 reveals a hydrophobic chamber to accommodate the acyl tails of lipids during the catalytic process and suggests a substrate-triggered product release mechanism. These results also provide insights into the catalytic mechanism of other eukaryotic enzymes in the CDP-AP family.

## Methods
### Protein expression and purification
The cDNA of human CEPT1 (NCBI reference sequence NM_006090.4) was cloned into the pCAG vector with an amino-terminal FLAG tag and a carboxyl-terminal 10× His tag. All the mutants were generated with a standard two-step PCR-based strategy. The primers for the mutants are listed in Supplementary Table 2. FreeStyle™ 293-F suspension cells

(Thermo Fisher Scientific) were cultured in SMM 293-TII (SinoBiological) at 37 °C and supplied with 5% $CO_2$. When the cell density reached $2.0 \times 10^6$ cells per mL, the cells were transiently transfected with the plasmids and polyethylenimines (Yeasen Biotechnology (Shanghai)). Approximately 1 mg of expression plasmids was premixed with 3 mg polyethylenimines in 50 ml fresh medium and incubated for 15–30 min before transfection. The 50 ml mixture was then added to 1 L of cell culture and incubated for 15–30 min. Transfected cells were cultured for 48 h before collection.

For purification of CEPT1, FreeStyle™ 293-F cells were collected and resuspended in buffer containing 25 mM Tris pH 8.0, 150 mM NaCl and protease inhibitor cocktails (Amresco). Then, the resuspended cells were solubilized with 1% (w/v) LMNG (Anatrace) at 4 °C for 2 h. After centrifugation at $20000 \times g$ for 1 h, the supernatant was collected and applied to anti-FLAG M2 resin (Sigma). The resin was rinsed with buffer A containing 25 mM Tris pH 8.0, 150 mM NaCl and 0.005% (w/v) GDN (Anatrace). The protein was eluted with buffer A plus 0.2 mg/ml FLAG peptide. Then, the eluate was loaded onto the NTA resin (Qiagen). After rinsing with buffer A, the protein was eluted with buffer A plus 250 mM imidazole. The eluate was concentrated and further purified by SEC (Superose 6 Increase 10/300 GL, Cytiva) in buffer A. The peak fractions were collected and concentrated to approximately 15 mg/ml for cryo sample preparation. The CEPT1 variants were purified using the same process described above.

## Enzymatic assays

To monitor the enzymatic activity of purified CEPT1 and its variants, 1 µl of 2.5 mg ml⁻¹ protein was diluted into a 19 µl solution containing micelles mixed with cholate, DAG (Sigma) and POPC (Anatrace), as described previously[24], with a final concentration of cholate at 0.4 mM, POPC at 2 mM and DAG at 2 mM in the reaction buffer (100 mM Tris pH 8.0, 20 mM $MgCl_2$, 1 mM EDTA). After incubation at 37 °C for two minutes, 1 µl of CDP-choline or CDP-ethanolamine was added to the indicated concentration to initiate the reaction. The mutational analyses and divalent ion dependence were performed in the presence of 0.5 mM CDP-choline.

Unless specifically mentioned, all reactions were allowed to proceed for 10 min at 37 °C. The reactions of CDP-choline were quenched by the addition of 1% SDS, followed by heating at 98 °C for 10 min, while those of CDP-ethanolamine were quenched by 1% SDS without heating. After centrifugation at $15,000 \times g$ for 5 min, 20 µl of the supernatant was subjected to HPLC (Agilent). Separation was carried out on an Eclipse XDB-C18 column (4.6 mm × 150 mm, 5 µm particle size, Agilent) with running buffer containing 10 mM tetrabutylammonium hydroxide, 10 mM $KH_2PO_4$ and 0.125% methanol, pH 7.00, and a flow rate of 1 ml/min at room temperature. The UV absorption at 267 nm was recorded to monitor the release of CMP. The peak position of CMP was identified using a standard sample (Sangon Biotech) and validated with mass spectrometry. Concentrations of released CMP were calculated by comparing peak areas with those of standard samples with known concentrations.

Three independent experiments were conducted, and a similar system without enzymes was applied to subtract the background noise for the wild type and variants. Nonlinear regression to the Michaelis–Menten equation was performed using GraphPad Prism 8.0.2.

## Cryo-EM sample preparation and data collection

The cryo grids were prepared using Thermo Fisher Vitrobot Mark IV. Quantifoil R1.2/1.3 Cu grids were glow-discharged with air for 60 s under 15 mA in PELCO easiGlow (Ted Pella). Aliquots of 3.5 µl of purified CEPT1, concentrated to approximately 15 mg/ml, were applied to glow-discharged grids. After blotting with filter paper for 3.5 s (100% humidity and 8 °C), the grids were plunged into liquid ethane cooled with liquid nitrogen. The CEPT1 and CDP-choline complex was reconstituted by incubating the purified enzyme (approximately 15 mg/ml) with 10 mM CDP-choline (Macklin) on ice for one hour, and then the cryo samples were prepared as described above.

The grid was loaded into a Titan Krios (FEI) electron microscope operating at 300 kV equipped with a BioQuantum energy filter and a K3 direct electron detector (Gatan). Images were automatically collected with EPU in the super-resolution mode. Defocus values varied from −1.5 to −2.0 µm. Image stacks were acquired with an exposure time of 4.5 s and fractionated into 32 frames with a total dose of 50 e⁻ Å⁻². The stacks were motion corrected with MotionCor2[30] and binned twofold, resulting in a pixel size of 1.07 Å/pixel, meanwhile dose weighting was performed[31]. The defocus values were estimated with CTFFIND4[32].

## Cryo-EM data processing

In total, 3116 and 1254 micrographs were collected for CEPT1 and its complex with CDP-choline, respectively.

For structural determination of CEPT1, a total of 6,982,016 particles were automatically picked with RELION[33–35]. After 2D classification, 851,097 particles were selected and subjected to ab initio reconstruction with 6 classes and C2 symmetry in cryoSPARC[36]. The good class with 297,380 particles was used as the seed for seed-facilitated classification[37], resulting in 2,545,220 good particles, which yielded a 3D reconstruction at approximately 4.3 Å. Then, these particles were further classified using three runs of heterogeneous refinement with multiple references and yielded a subset of 771,881 particles resulting in a 3D reconstruction at 4.0 Å. Eventually, the particles were applied into an ab initio reconstruction with 5 classes and C2 symmetry, and a subset of 459,021 particles was selected, which resulted in a 3D reconstruction with an overall resolution of 3.7 Å. These particles also resulted in a similar 3D reconstruction at 3.9 Å with C1 symmetry, supporting the application of C2 symmetry during data processing.

For structural determination of the CDP-choline complex, a total of 2.254.258 particles were automatically picked with RELION. After one run of heterogeneous refinement, 476,898 particles were selected and subjected to ab initio reconstruction with 3 classes and C2 symmetry in cryoSPARC. The good class with 215,057 particles was used as the seed for seed-facilitated classification, and a subset of 743,393 particles was selected and yielded a 3D reconstruction with 4.7 Å. After two runs of heterogeneous refinement with multiple references and one run of ab initio reconstruction, a subset of 249,493 particles was eventually selected to yield a 3D reconstruction at 3.8 Å.

Resolutions were estimated with the gold-standard Fourier shell correlation 0.143 criterion[38] with high-resolution noise substitution[39]. The local resolution maps were calculated using "Local Resolution Estimation" in cryoSPARC.

## Model building and refinement

The 3.7 Å map of CEPT1 was initially used for model building. The CEPT1 model predicted by AlphaFold2[26] was used as the initial model to be docked into the map in Chimera[40]. Manual adjustment was then made in Coot[41] to generate the final structure. Then, the structure was fitted into the complex map of CEPT1 and CDP-choline. All structure refinement was carried out by PHENIX[42] in real space with secondary structure and geometry restraints. Overfitting of the models was monitored by refining the model in one of the two independent maps from the gold-standard refinement approach and testing the refined model against the other map[43]. All structural figures were prepared with PyMol[44].

## Statistics and reproducibility

No statistical methods were used to predetermine the sample size. The experiments were not randomized, and the investigators were not blinded to allocation during experiments and outcome assessment. Each experiment was conducted independently at least twice with similar results.

## Reporting summary

Further information on research design is available in the Nature Portfolio Reporting Summary linked to this article.

## Data availability

The atomic coordinates of CEPT1 and its complex with CDP-choline complexes have been deposited in the RCSB Protein Data Bank under the accession codes 8GYX and 8GYW. The corresponding electron microscopy maps have been deposited in the Electron Microscopy Data Bank under the accession codes EMD-34379 and EMD-34378, respectively. The raw electron microscopy images used to build the 3D structure are available from the corresponding author (hongwuqian@ustc.edu.cn) upon request. The paper makes use of RCSB Protein Data Bank accession codes 4MND, 6WM5, and 7B1K. Source data are provided with this paper.

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

## Acknowledgements

We thank the Cryo-EM Center of the University of Science and Technology of China (USTC) for the EM facility support. We are grateful to Dr. Yong-Xiang Gao and all the other staff members at the Cryo-EM Center for their technical support on cryo-EM data collection. We thank Wen-Tao Hou and Yong-Liang Jiang for their technical support with the HPLC assays. This work was supported by the National Natural Science Foundation of China (32271241), the Fundamental Research Funds for the Central Universities (WK9100000031), "the Talent Fund Project of Biomedical Sciences and Health Laboratory of Anhui Province, University of Science and Technology of China" (BJ9100000003), and the start-up funding from the University of Science and Technology of China (KY9100000034 and KJ2070000082).

## Author contributions

H.Q. conceived the project and designed the experiments. Z.W., M.Y., Y.H. and Y.Y. performed cloning and protein purification. M.Y. and Z.W. prepared cryo-EM samples and collected data. H.Q. determined the structures. Z.W. and Y.Y. performed the HPLC-based activity assays. All authors contributed to the data analysis and preparation of the manuscript. H.Q. wrote the manuscript.

## Competing interests

The authors declare no competing interests.
