## [Peer Review File · Nature Communications]

Making membranes: Structure of human choline/ ethanolamine phosphotransferase 1Reviewer #1 (Remarks to the Author):

In this work the integral membrane CEPT1 protein, which catalyzes the final step in the synthesis of phosphatidylcholine (PC) and phosphatidylethanolamine (PE) - the two major membrane phospholipids found in cells - was determined by cryoEM. The CEPT1 protein (with FLAG and poly-His tags at the N-terminus) was purified after expression in FreeStyle293-F cells. Purification required solubilization of membranes with detergent and stepwise purification using anti-FLAG resin, NTA resin, and then size exclusion via a Superose column. CEPT1 variants were purified using a similar protocol. The purified CEPT1 protein was studied by cryo-EM to determine structure using the AlphaFold2 predicted structure as a docking guide. The original structure appears to contain the product, PC. The final resolved structure was at 3.7 angstroms. A structure of CEPT1 was also resolved after incubation with the substrate CDP-choline and the presence of CDP-choline in the active site resulted in the exclusion of the product, PC, within the cryoEM structure. Structure-activity relationships were probed by site-directed mutagenesis variants and enzyme activity assays and further refinement of substrate/product binding was determined resulting in a catalytic mechanism was proposed. This is the first experimentally resolved structure for CEPT1 and many new molecular aspects by which the major membrane phospholipids are synthesized were described.

Comments

- 1. I would suggest a paragraph on the information added by this study beyond that which could be surmised from the AlphaFold2 structure itself for CEPT1**
- 2. Suppl Fig 9c - "CDP-coline" should be "CDP-choline" in the actual Figure itself**
- 3. I might suggest a title with more gravitas, perhaps along the lines of "Making Membranes: Structure of human choline/ethanolamine phosphotransferase 1" or something similar**

Reviewer #2 (Remarks to the Author):

This paper provides the first structure of a key enzyme of mammalian phosphatidylcholine/phosphatidylethanolamine synthesis and makes a significant contribution to the field. PC and PE are the major phospholipids of the cell and thus their structure provides a mechanism whereby a hydrophilic molecule i.e. CDP-choline from the cytosol and diacylglycerol, a hydrophobic substrate are combined to make the phospholipid, PC. The enzyme is characterised with respect to its enzymatic properties including dependence on Mg²⁺.

Comments

I am puzzled by the reference to the alkyl chains of the PC molecule. The majority of PC in cells is PC with acyl chains. But throughout this paper, reference is made to alkyl chains only. Does the enzyme discriminate between acyl chains and alkyl chains? This needs to be clarified.

Reviewer #3 (Remarks to the Author):

Manuscript:

Wang et al., "Structural basis for catalysis of human choline/ethanolamine phosphotransferase 1 in the Kennedy pathway"

Summary:

Wang et al. report structures of a class II CDP-alcohol phosphotransferase, human CEPT1 in states bound to the lipid product and to the donor substrate, CDP-choline. The structures show the lipid product (PC) with alkyl tails bound in a hydrophobic pocket within the transmembrane region, and the authors suggest that the absence of the

product in the donor-bound state suggests a substrate-triggered product release mechanism. The authors also identify a dimerization motif in the eukaryotic class II CDP-APs which may have resulted from an internal duplication event.

Main Impressions:

Overall, this is a well conducted study, well written, and represents an important milestone for the field – these structures will be important in understanding the mechanism of lipid biosynthesis in eukaryotes. I have only a few specific comments to address, itemized below.

Specific Comments:

- The clash scores for the structures are vastly higher than is acceptable – greater than 60 for both structures. Automatic real space refinement with PHENIX brings them down to more reasonable (but still high) values of 13-15, so I suspect a technical error or oversight of some kind. This must be addressed.
- The authors state “A representative 2D average indicates that hCEPT1 may be purified as a dimer... therefore, C2 symmetry was applied during data processing.”. It is not reasonable to conclude that an object has C2 symmetry based on a single 2D class. At a minimum, the authors should perform C1 ab initio reconstruction and C1 refinement, to verify that the dimer is in fact C2 symmetric – asymmetric or pseudosymmetric dimers are entirely possible (it is also possible that there is a mixture, if occupancy of the ligand binding sites is partial). It would also be worth considering performing C2 symmetry expansion, followed by local refinement with a mask around a single protomer, in order to improve resolution in the ligand binding sites.
- Labeling of many figures is low contrast and hard to read. For example, in Ext. Data Fig 4 a, labelling the domain diagram with dark cyan text on a slightly lighter cyan background. I would suggest increasing contrast and improving readability where possible. Also in this figure, I would suggest trying some different approaches to show better the C2 pseudo-symmetry of the internal repeat. When I look at the structure in Chimera, it is apparent to me, but the representation in E.D. 4e does not show it clearly. I also wonder whether a structure-based sequence alignment of the two regions might be informative.
- An unusual feature of the structure is the group of 4 histidine residues protruding into the cleft between the dimers, approximately halfway across the membrane. These seem to be almost close enough together for an ion binding site, and the authors note some unmodeled density in this region. Can the authors add a sentence or two giving their interpretation of this unusual feature? How does this affect the thickness of the membrane in this region?
- The idea of substrate-triggered product release is intriguing, but I would be a bit more cautious about this – I am not sure that what happens in such a vast excess of the donor substrate (10mM) is necessarily representative of what would happen under physiological conditions.
- On Page 4, line 56, the authors refer to “a lipophilic CDP-linked donor and a polar receptor” – I think “acceptor” is intended here?

Point-by-point responses:

Reviewer #1:

This reviewer recognized the significance of our study and commented “..., *many new molecular aspects by which the major membrane phospholipids are synthesized were described*”. The specific comments are addressed below.

1. I would suggest a paragraph on the information added by this study beyond that which could be surmised from the AlphaFold2 structure itself for CEPT1

We thank the reviewer for the suggestion. Two sentences were added on Page 12 in the revised manuscript: “The CEPT1 structure is a dimer, which is distinct from the monomeric model predicted by AlphaFold2. Moreover, our structures exhibit the coordination of product PC and substrate CDP-choline in hCEPT1”.

2. Suppl Fig 9c – “CDP-coline” should be “CDP-choline” in the actual Figure itself.

Point taken. We have corrected this in the revised version.

3. I might suggest a title with more gravitas, perhaps along the lines of “Making Membranes: Structure of human choline/ethanolamine phosphotransferase 1” or something similar

Thanks for the suggestion. We have changed the title according to the reviewer’s suggestion.

We thank the reviewer for these constructive comments.

Reviewer #2

Reviewer #2 recognized the significance of our study and commented that our study “*makes a significant contribution to the field*”. He/She had a minor puzzler about our manuscript, which is addressed below.

I am puzzled by the reference to the alkyl chains of the PC molecule. The majority of PC in cells is PC with acyl chains. But throughout this paper, reference is made to alkyl chains only. Does the enzyme discriminate between acyl chains and alkyl chains? This needs to be clarified.

We apologize for using the incorrect description, which confused the reviewer. “Acyl” should be correct. We have corrected “*alkyl*” to “*acyl*” in our revised manuscript.

We thank the reviewer for the comments.

Reviewer #3:

This reviewer commented that *“this is a well conducted study, well written, and represents an important milestone for the field – these structures will be important in understanding the mechanism of lipid biosynthesis in eukaryotes”*. He/She raised a few specific comments which are addressed below.

Specific Comments:

The clash scores for the structures are vastly higher than is acceptable – greater than 60 for both structures. Automatic real space refinement with PHENIX brings them down to more reasonable (but still high) values of 13-15, so I suspect a technical error or oversight of some kind. This must be addressed.

We thank the reviewer for pointing out these poor values. We refined our models carefully, and the clash scores were improved to 6.23 and 6.64, respectively. The new validation files were updated in the revision.

The authors state “A representative 2D average indicates that hCEPT1 may be purified as a dimer... therefore, C2 symmetry was applied during data processing.”. It is not reasonable to conclude that an object has C2 symmetry based on a single 2D class. At a minimum, the authors should perform C1 ab initio reconstruction and C1 refinement, to verify that the dimer is in fact C2 symmetric – asymmetric or pseudosymmetric dimers are entirely possible (it is also possible that there is a mixture, if occupancy of the ligand binding sites is partial). It would also be worth considering performing C2 symmetry expansion, followed by local refinement with a mask around a single protomer, in order to improve resolution in the ligand binding sites.

Thanks for this critical comment. C1 was applied at the beginning of data processing. However, we cannot obtain a reasonable initial model when using C1 symmetry. We then tried C2 symmetry and fortunately obtained a good initial model. Therefore, we kept C2 symmetry in the following data processing process. To address the reviewer’s concerns, we applied C1 refinement to the final good particles and obtained a reconstruction at 3.9 Å. Map comparison suggests that this map is almost identical to our previous C2 map except for poor resolution. The final map in Extended Data Fig. 3a was updated to compare two maps. We also added one sentence to describe this issue in the revised manuscript---“These particles also resulted in a similar 3D map at a slightly poor resolution when applying C1 symmetry, supporting the application of C2 symmetry (Extended Data Fig. 3a)” (Page 6, the end of 2st paragraph)

We also tried C2 symmetry expansion before, but we didn't get any improvements.

Labeling of many figures is low contrast and hard to read. For example, in Ext. Data Fig 4 a, labelling the domain diagram with dark cyan text on a slightly lighter cyan background. I would suggest increasing contrast and improving readability where possible. Also in this figure, I would suggest trying some different approaches to show better the C2 pseudo-symmetry of the internal repeat. When I look at the structure in Chimera, it is apparent to me, but the representation in E.D. 4e does not show it clearly. I also wonder whether a structure-based sequence alignment of the two regions might be informative.

Points taken. The contrast of the labels in Ext. Data Fig 4a was increased, and the Ext. Data Fig 4e was remade to show the C2 pseudosymmetry clearly. Please check the updated Ext. Data Fig 4 for details. We also applied the structure-based sequence alignment of the catalytic domain and dimerization domain as suggested by the reviewer (results are shown below). These two regions are not conserved, and we didn't get more informative information.

An unusual feature of the structure is the group of 4 histidine residues protruding into the cleft between the dimers, approximately halfway across the membrane. These seem to be almost close enough together for an ion binding site, and the authors note some unmodeled density in this region. Can the authors add a sentence or two giving their interpretation of this unusual feature? How does this affect the thickness of the membrane in this region?

We thank the reviewer for this suggestion. Indeed, the four histidine residues exhibit an unusual feature. However, there is no evidence to support that they may coordinate an ion. Because the histidine residues are close to the extra densities in the interface, we proposed that they may help stabilize the dimerization through interaction with the polar head of amphiphilic molecules at the dimeric interface. This structural feature also leaves a cleft between two

protomers. It seems that the membrane should be thinner at this region. We introduced mutations to these residues, but the catalytic activities were not influenced. The physiological role of unusual structural features is still unknown. Anyway, we rewrote the part in the main text to describe the cluster of histidine residues as follows (Page 7, 2nd paragraph).

“However, some extra densities were sandwiched by hydrophobic residues and a cluster of four histidine residues from two protomers at the cytosolic leaflet (Extended Data Fig. 5a). This observation indicates that some amphipathic molecules (endogenous lipids or exogenous detergents) may be involved in dimer formation, whose nonpolar tails and polar heads may be stabilized by hydrophobic residues and histidine residues, respectively”

The idea of substrate-triggered product release is intriguing, but I would be a bit more cautious about this – I am not sure that what happens in such a vast excess of the donor substrate (10mM) is necessarily representative of what would happen under physiological conditions.

We thank the reviewer for this critical comment. We agree with the reviewer’s concern and tried to set biochemical assays to support the substrate-triggered product release mechanism. Unfortunately, all trials failed. We therefore weakened our description about this model in the main text and added one sentence to describe this concern in the discussion---“Because the complex structure of CEPT1 and CDP-choline were solved at the condition with vast excess of substrate (10 mM CDP-choline vs 0.3 mM enzymes), this speculation requires biochemical validation under physiological conditions in the future” (Page 12, 2nd paragraph).

On Page 4, line 56, the authors refer to “a lipophilic CDP-linked donor and a polar receptor” – I think “acceptor” is intended here?

We thank the reviewer for pointing out this error. We have corrected them in the revised manuscript.

We thank the reviewer for these critical and constructive comments.

Reviewer #2 (Remarks to the Author):

The authors have made the changes as suggested.

Reviewer #3 (Remarks to the Author):

I thank the authors for their careful and comprehensive response to my comments. I have no further additions regarding the revised version of the manuscript.